# The coral pathogen *Vibrio coralliilyticus* uses a T6SS to secrete a group of novel anti-eukaryotic effectors that contribute to virulence

Shir Mass[1], Hadar Cohen[1], Ram Podicheti[2], Douglas B. Rusch[2], Motti Gerlic[1], Blake Ushijima[3], Julia C. van Kessel[4], Eran Bosis[5], Dor Salomon[1]*

1 Department of Clinical Microbiology and Immunology, School of Medicine, Faculty of Medical and Health Sciences, Tel Aviv University, Tel Aviv, Israel, 2 Center for Genomics and Bioinformatics Indiana University, Bloomington, Indiana, United States of America, 3 Department of Biology and Marine Biology, University of North Carolina Wilmington, Wilmington, North Carolina, United States of America, 4 Department of Biology, Indiana University, Bloomington, Indiana, United States of America, 5 Department of Biotechnology Engineering, Braude College of Engineering, Karmiel, Israel

* dorsalomon@mail.tau.ac.il

**Data Availability Statement:** Relevant data are within the paper and its Supporting Information

## Abstract

*Vibrio coralliilyticus* is a pathogen of coral and shellfish, leading to devastating economic and ecological consequences worldwide. Although rising ocean temperatures correlate with increased *V. coralliilyticus* pathogenicity, the specific molecular mechanisms and determinants contributing to virulence remain poorly understood. Here, we systematically analyzed the type VI secretion system (T6SS), a contact-dependent toxin delivery apparatus, in *V. coralliilyticus*. We identified 2 omnipresent T6SSs that are activated at temperatures in which *V. coralliilyticus* becomes virulent; T6SS1 is an antibacterial system mediating interbacterial competition, whereas T6SS2 mediates anti-eukaryotic toxicity and contributes to mortality during infection of an aquatic model organism, *Artemia salina*. Using comparative proteomics, we identified the T6SS1 and T6SS2 toxin arsenals of 3 *V. coralliilyticus* strains with distinct disease etiologies. Remarkably, T6SS2 secretes at least 9 novel anti-eukaryotic toxins comprising core and accessory repertoires. We propose that T6SSs differently contribute to *V. coralliilyticus*'s virulence: T6SS2 plays a direct role by targeting the host, while T6SS1 plays an indirect role by eliminating competitors.

## Introduction

The oceans are home to gram-negative marine bacteria of the genus *Vibrio*. These include many established and emerging pathogens that infect humans and marine animals [1,2]. In the past, human pathogenic vibrios were primarily associated with the warmer equatorial waters. Yet, in recent decades, they have spread to other regions, including the northern United States of America, Canada, and North Europe [3,4]. This spread correlates with rising ocean surface-level temperatures and disease outbreaks [5,6].

files. The mass spectrometry proteomics data have been deposited in the ProteomeXchange Consortium with the dataset identifier PXD049479.

**Funding:** This project received funding from the National Science Foundation (https://www.nsf.gov) and US-Israel Binational Science Foundation (https://www.bsf.org.il) (NSF-BSF) under award numbers IOS-2207168 (JCVK), IOS-2207169 (BU), and 2021733 (DS), and from the Israel Science Foundation (www.isf.org.il; ISF grant number 1362/21 to DS and EB, and grant number 2174/22 to MG). The funders played no role in the study design, data collection and analysis, decision to publish, or preparation of the manuscript.

**Competing interests:** The authors have declared that no competing interests exist.

**Abbreviations:** BMDM, bone marrow-derived macrophage; CDS, coding sequences; CFU, colony-forming unit; CoVe, Coralliilyticus virulence effector; CNF, cytotoxic necrotizing factor; ECL, enhanced chemiluminescence; eGFP, enhanced GFP; FDR, false discovery rate; GASW, glycerol artificial seawater; LB, lysogeny broth; LFQ, Label-free quantification; MCS, multiple cloning site; MLB, marine lysogeny broth; MOI, multiplicity of infection; PI, propidium iodide; T6SS, type VI secretion system; WT, wild-type.

Corals are marine animals affected by rising ocean temperatures caused by climate change and the spread of vibrios [7–9]. They are ecologically and economically important because they provide diverse ecosystems used as habitats for various fish and invertebrates, as well as help to protect shorelines from storm surges and erosion [10]. The coral animal lives in a symbiotic relationship with photosynthetic endosymbiotic dinoflagellates and microbes (collectively called the coral holobiont) [11–14]. *Vibrio coralliilyticus* is a bacterial pathogen shown to be a cause of diseases resulting in bleaching or tissue loss in corals [9,15,16]. Among other coral pathogens [9], *V. coralliilyticus* stands out due to its wide geographic spread and broad range of reported hosts. Aside from corals, *V. coralliilyticus* is also responsible for mortalities in shellfish hatcheries [17].

The coral holobiont is affected by various environmental conditions, such as shifts in water temperature, pH, and nutrients. Elevated temperature is a key factor in many *V. coralliilyticus* infections because it increases the abundance and virulence of many *V. coralliilyticus* strains [14]. At temperatures below 23˚C, *V. coralliilyticus* strains are predominantly not pathogenic [8]. However, the virulence of many strains increases when temperatures rise above 23˚C [14,15,18]. In some cases, the symbiotic dinoflagellates are killed and coral bleaching occurs. With most pathogenic strains, shifts to >27˚C result in coral tissue lysis and increased coral mortality [15]. Elevated temperatures are associated with the production of proteases and hemolysins, motility, antimicrobial resistance, and secretion systems in *V. coralliilyticus* [19]. In addition, the expression of *toxR*, a transcription regulator associated with virulence in other vibrios [20], correlates with increased temperature and was shown to contribute to *V. coralliilyticus* virulence [21]. These data provide strong evidence that temperature regulates virulence-associated genes in *V. coralliilyticus*. Nevertheless, it remains unclear how these factors contribute to pathogenicity and whether the same factors play a role in virulence towards different hosts.

Many vibrios employ a specialized toxin delivery mechanism, the type VI secretion system (T6SS), to manipulate their environment [22–29]. The T6SS is a proteinaceous apparatus that is assembled inside the bacterial cell: a sheath structure engulfs an inner tube made of stacked hexameric rings of Hcp proteins, which is capped by a spike comprising a VgrG trimer sharpened by a PAAR repeat-containing protein (hereafter referred to as PAAR) [30]. This tube-spike complex is decorated with toxic proteins, called effectors, that mediate the toxic activities of the T6SS [31–33]. Contraction of the sheath propels the tube-spike complex out of the cell, providing it with sufficient force to penetrate the membrane of a neighboring cell where effectors are deployed [34]. Whereas most T6SSs investigated to date mediate interbacterial competitions by delivering antibacterial effectors, a few T6SSs have been shown to target eukaryotes and mediate virulence [33,35,36]. In accordance, although most *Vibrio* T6SSs play a role in interbacterial competitions [24–28,37–39], we and others recently revealed *Vibrio* T6SSs and effectors that target eukaryotes, and we postulated that they play a role in virulence [22,40–45].

Several studies reported the temperature-dependent expression of T6SS components in *V. coralliilyticus* [19,42], suggesting that T6SSs play a role in the temperature-regulated transition to a pathogenic lifestyle. The antibacterial activity of one T6SS was previously demonstrated in 2 *V. coralliilyticus* strains [42,46]. However, the presence of other T6SSs in the *V. coralliilyticus* pan-genome, their role, regulation, effector repertoire, and contribution to virulence remain unknown. Here, we systematically analyzed the T6SSs in the *V. coralliilyticus* pan-genome and revealed 2 omnipresent systems. Using 3 *V. coralliilyticus* strains as model systems, we experimentally defined the environmental conditions regulating the activation of these 2 T6SSs. We also identified their function and effector repertoires. Importantly, we revealed 9 novel anti-eukaryotic effectors delivered by the *V. coralliilyticus* T6SS2, contributing to *V. coralliilyticus* virulence.

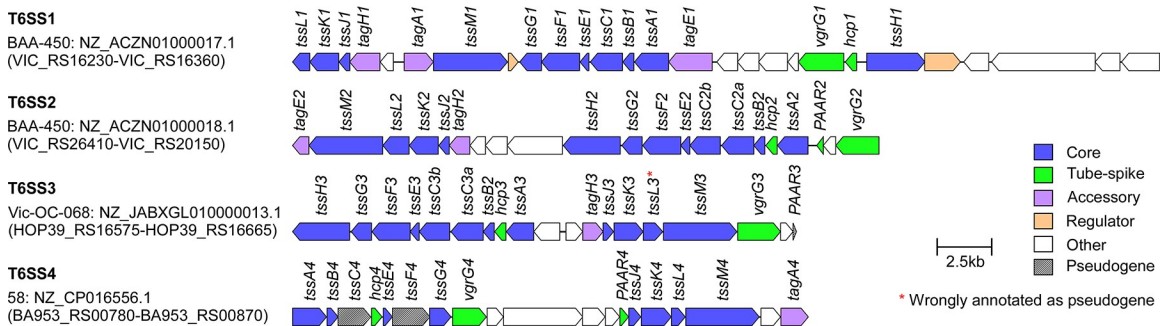

**Fig 1. Representative T6SS gene clusters found in *Vibrio coralliilyticus* genomes.** The strain name, GenBank accession number, and the first and last locus tag are denoted on the left. Genes are denoted by arrows indicating the predicted direction of transcription. Encoded proteins or domains are denoted above the genes. T6SS, type VI secretion system.

## Results

### Two T6SSs are omnipresent in *Vibrio coralliilyticus* strains

To identify the T6SSs found in the pan-genome of *V. coralliilyticus*, we retrieved the sequences of the core T6SS sheath component, TssB, from 31 available RefSeq *V. coralliilyticus* genomes (**S1 Dataset**) and analyzed their genomic neighborhoods. Our analyses revealed that all genomes harbor 2 conserved T6SSs, named T6SS1 and T6SS2 (**Fig 1** and **S2 Dataset**), suggesting that these T6SSs play an important role in the *V. coralliilyticus* lifestyle. T6SS1 is similar to the previously investigated T6SS1 from *V. parahaemolyticus* [24,47], *V. alginolyticus* [26], and *V. proteolyticus* [28], sharing the same gene content and organization. We recently showed that this system mediates interbacterial competition in the *V. coralliilyticus* type strain BAA-450 and in strain OCN008 [42,46]. The synteny of the T6SS1 (**Fig A in S1 Text**) and T6SS2 (**Fig B in S1 Text**) gene clusters is similar in all strains, with the exception of the 5′ region of T6SS1 containing different versions of a predicted effector, co-effector, and immunity protein combinations [48], as previously observed in similar T6SS gene clusters in *V. parahaemolyticus* [49,50]. Two additional T6SSs, which we named T6SS3 and T6SS4, are each found in a single *V. coralliilyticus* genome (**S2 Dataset**). Notably, 2 genes encoding structural core components in T6SS4 appear to include frameshifts, and the gene cluster lacks a gene encoding the conserved T6SS core component, TssH (**Fig 1**). Therefore, it is possible that T6SS4 is not functional.

### Environmental conditions regulate *Vibrio coralliilyticus* T6SSs

Because T6SS1 and T6SS2 are omnipresent in *V. coralliilyticus*, we set out to investigate their activation and function. First, we sought to determine whether T6SS1 and T6SS2 are regulated by environmental conditions regulating *V. coralliilyticus* virulence. To this end, we selected 3 representative *V. coralliilyticus* strains harboring both T6SSs: BAA-450 (the type strain), OCN008, and OCN014. These strains were isolated from different coral hosts and display different disease etiologies [16,51,52]. Strains BAA-450 and OCN014 have a temperature-dependent infection mode, and they become more virulent as temperatures rise above 23°C; the virulence of strain OCN008 does not significantly change from 23 to 27°C [16,18,21].

To determine whether the activation of T6SS1 and T6SS2 depends on temperature or nutrient availability, we monitored the expression and secretion of the conserved secreted T6SS structural components, VgrG1 and Hcp2 [23], respectively. Bacteria were grown in either rich (LB containing 3% [wt/vol] NaCl; MLB) or poor (glycerol artificial seawater; GASW) media

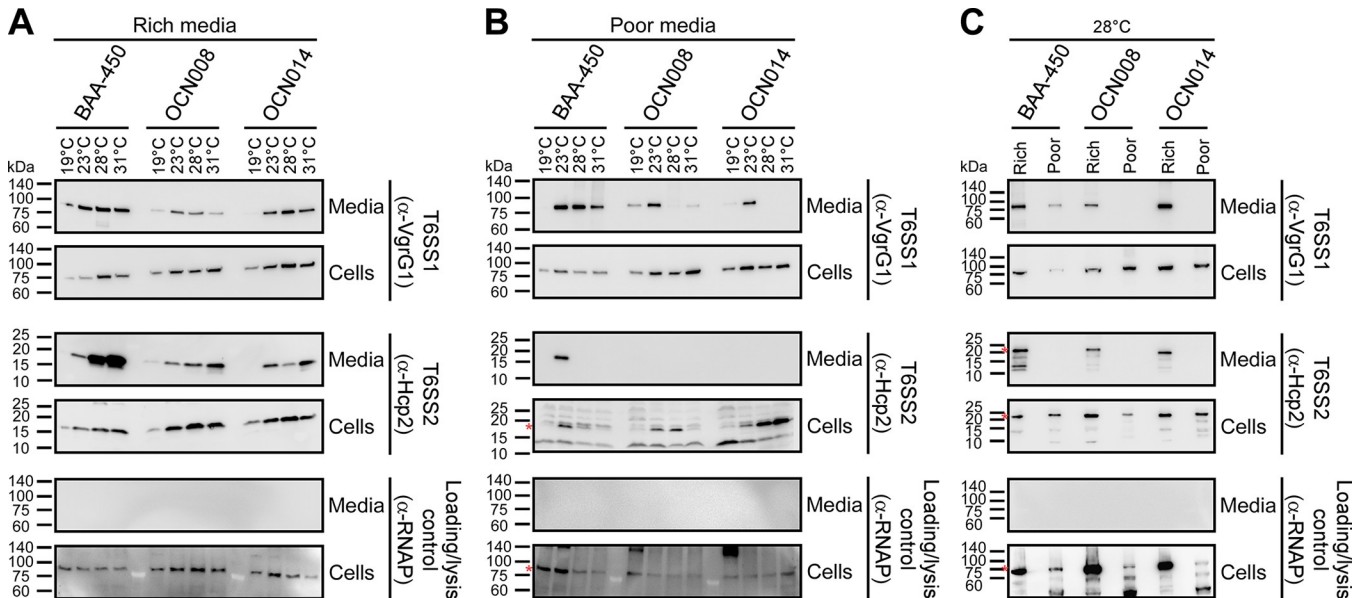

**Fig 2. *Vibrio coralliilyticus* T6SS1 and T6SS2 are regulated by environmental conditions.** Expression (cells) and secretion (media) of VgrG1 and Hcp2 from the 3 indicated *V. coralliilyticus* strains grown for 4 h at the indicated temperatures in "rich" LB containing 3% [wt/vol] NaCl (MLB) **(A)** or "poor" glycerol artificial seawater medium **(B)**. **(C)** Comparison of VgrG1 and Hcp2 expression and secretion when *V. coralliilyticus* strains were grown at 28°C in "rich" or "poor" media. RNA polymerase sigma 70 (RNAp) was used as a loading and lysis control. Asterisks denote expected protein sizes. Results from a representative experiment out of at least 3 independent experiments are shown. LB, lysogeny broth; T6SS, type VI secretion system.

and under a range of physiologically relevant temperatures that affect *V. coralliilyticus* pathogenicity: 19, 23, 28, and 31°C [16,21,51]. As shown in **Fig 2A and 2B**, we found that the activity of both T6SS1 and T6SS2 is temperature and media dependent. In rich media, both systems are active between 23 and 31°C; T6SS1 secretion peaks at 28°C, whereas T6SS2 secretion peaks at 31°C (**Fig 2A**). Notably, secretion via T6SS1 in strain OCN008 appears lower than in BAA-450 and OCN014. In poor media, T6SS1 secretion peaks at 23°C in all strains and is retained at higher temperatures only in strain BAA-450 (**Fig 2B**); T6SS2 secretion is only observed in strain BAA-450 at 23°C. Comparison between the activity of both systems in rich and poor media at 28°C revealed higher levels of expression and secretion in rich media (**Fig 2C**). Therefore, unless otherwise indicated, we performed subsequent analyses of T6SS1 and T6SS2 when *V. coralliilyticus* strains are grown in rich media at 28°C, conditions in which both systems are active in all 3 strains.

## T6SS1 mediates interbacterial competitions

We previously reported that T6SS1 in strains BAA-450 and OCN008 mediates antibacterial activity during interbacterial competitions [42,46]. To determine whether this is also true for T6SS1 in strain OCN014 and whether T6SS2 also plays a role in interbacterial competition, we set out to monitor the outcome of interbacterial competitions using *V. coralliilyticus* strains in which the 2 T6SSs were inactivated, either individually or together. To this end, we first constructed *V. coralliilyticus* mutant strains in which we inactivated T6SS1 by deleting the gene encoding the conserved structural component Hcp1 (Δ*hcp1*) and T6SS2 by deleting the gene encoding the conserved structural component TssM2 (Δ*tssM2*) (**Fig C in S1 Text, panel A**). These mutations did not affect bacterial growth (**Fig C in S1 Text, panel B**). When competed against a sensitive *V. natriegens* prey strain on rich media plates at 28°C, all 3 *V. coralliilyticus* strains killed the prey, evident by the decrease in prey viability during the 4 h of co-incubation

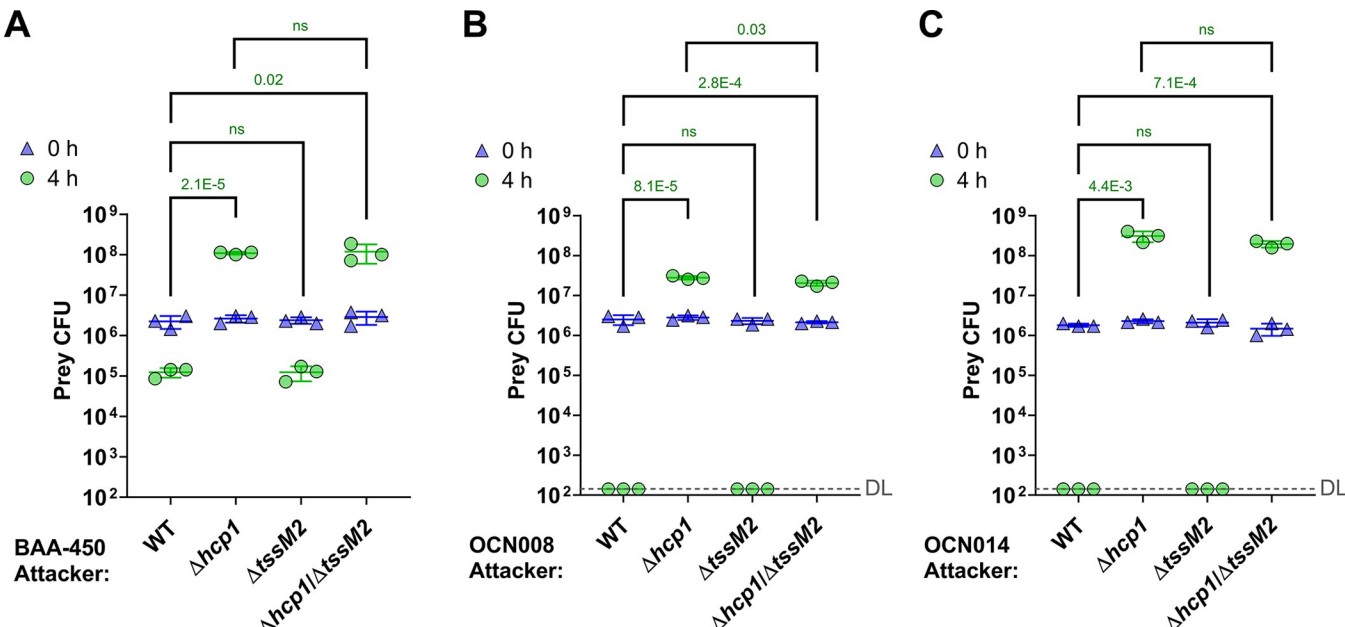

**Fig 3. *Vibrio coralliilyticus* T6SS1 mediates interbacterial competition. (A–C)** Viability counts (CFU) of *V. natriegens* prey strains before (0 h) and after (4 h) co-incubation with the indicated *V. coralliilyticus* BAA-450 (A), OCN008 (B), or OCN014 (C) attacker strains on MLB plates at 28°C. The statistical significance between samples at the 4 h time point was calculated using an unpaired, two-tailed Student's *t* test; ns, no significant difference (*P* > 0.05); WT, wild-type; DL, the assay's detection limit. Data are shown as the mean ± SD; *n* = 3. The data shown are a representative experiment out of at least 3 independent experiments. The data underlying this figure can be found in S1 Data. CFU, colony-forming unit; T6SS, type VI secretion system.

with the wild-type *V. coralliilyticus* attackers (**Fig 3**). This killing was dependent on T6SS1, since its inactivation in the attacker strains by deleting *hcp1* abolished the toxicity. The T6SS1-mediated killing was also apparent when *V. coralliilyticus* OCN008 was competed against *V. alginolyticus* or *V. campbellii*, 2 species that were previously isolated from corals [53] (**Fig D in S1 Text**). Inactivation of T6SS2 by deleting *tssM2*, either alone or in combination with an inactive T6SS1, had no effect on the observed antibacterial activity of *V. coralliilyticus* (**Fig 3**). Taken together, our results confirm that the *V. coralliilyticus* T6SS1 mediates antibacterial activity and suggest that T6SS2 does not play a role in interbacterial competition.

## T6SS2 targets eukaryotes

Based on the above results, we hypothesized that T6SS2 mediates anti-eukaryotic activities. To investigate whether T6SS2 plays a role in bacterial virulence, we employed the saline lake-dwelling brine shrimp, *Artemia salina*, as an aquatic animal model [41,54,55]. Wild-type *V. coralliilyticus* OCN008 was lethal to *Artemia* nauplii (larvae), with a median survival of 53 h. Inactivation of T6SS2, either alone (Δ*tssM2*) or together with T6SS1 (Δ*hcp1*/Δ*tssM2*), resulted in a significantly reduced lethality (median survival undefined or 56 h, respectively), whereas inactivation of T6SS1 (Δ*hcp1*) had no effect (**Fig 4A**). These results reveal a role for the *V. coralliilyticus* T6SS2 in pathogenicity during infection of a eukaryotic host.

To further investigate the anti-eukaryotic activity of *V. coralliilyticus* T6SS2 in all 3 strains, we used real-time microscopy to monitor *V. coralliilyticus*-mediated cell death kinetics. To this end, we employed bone marrow-derived macrophages (BMDMs), which have been previously used as a model to monitor the toxic effects of another *Vibrio* T6SS [40]. Various levels of cell death were observed starting ~30 min after adding either of the wild-type *V. coralliilyticus* strains BAA-450, OCN008, or OCN014 (**Fig 4B–4D**). Remarkably, inactivation of T6SS2,

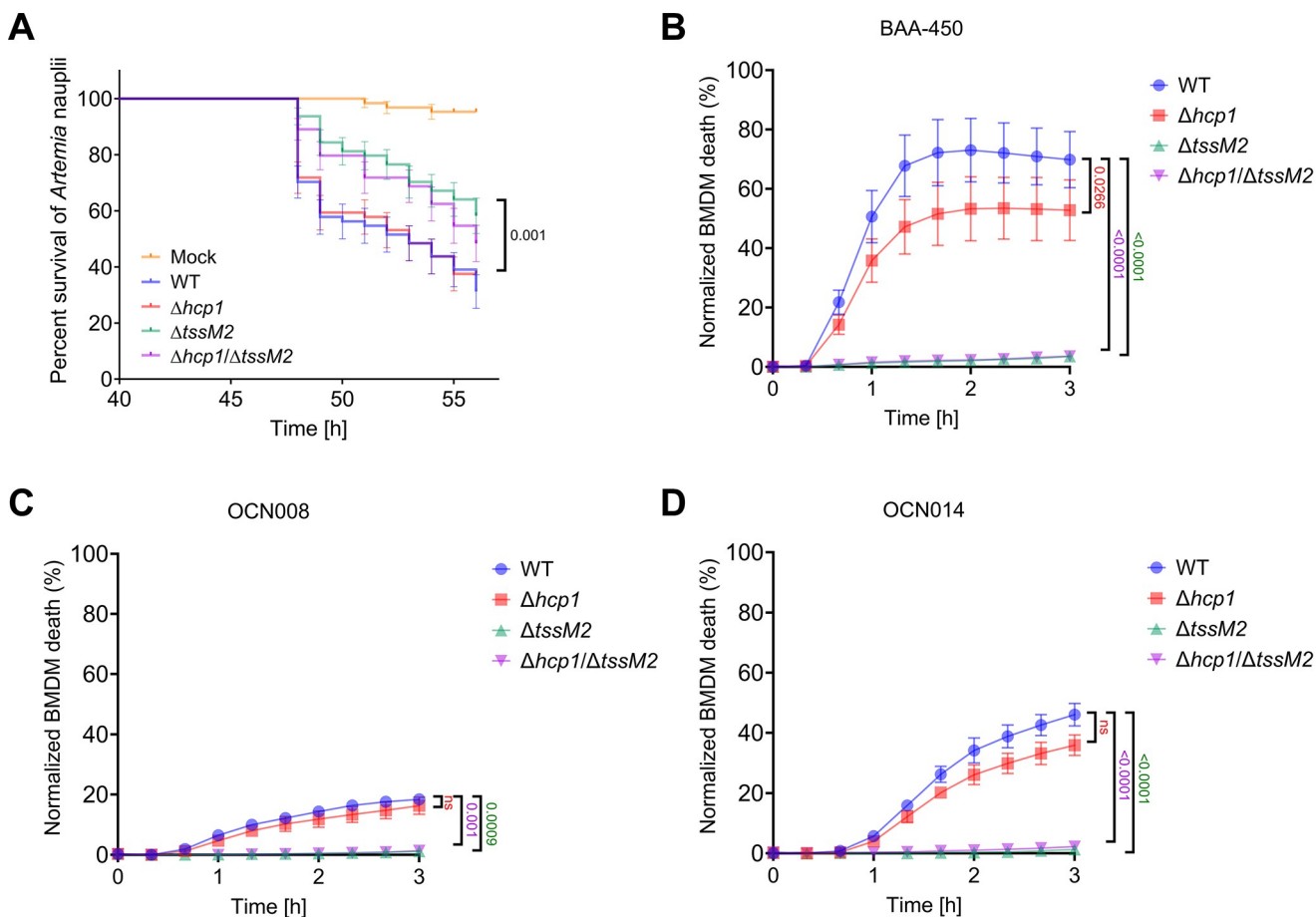

**Fig 4. *Vibrio coralliilyticus* T6SS2 mediates lethality in *Artemia* nauplii and in macrophages. (A)** *Artemia* nauplii were challenged with the indicated *V. coralliilyticus* OCN008 strains, and survival was assessed 40 to 56 h postinfection. Approximately $5 \times 10^7$ bacteria were added to each well containing 2 nauplii. Data are shown as the mean ± SE of 4 biological replicates, each comprising 16 nauplii for every bacterial strain. The statistical significance between the WT and Δ*tssM2* curves was calculated using the Log-rank (Mantel–Cox) test. **(B–D)** Assessment of cell death upon infection of BMDMs with the indicated *V. coralliilyticus* BAA-450 (B), OCN008 (C), or OCN014 (D) strains. Approximately $3.5 \times 10^4$ BMDMs were seeded into 96-well plates in triplicates and infected with *V. coralliilyticus* strains at an MOI ~ 4. PI was added to the medium prior to infection, and its uptake kinetics were assessed using real-time microscopy. WT, wild-type. Results from a representative experiment out of at least 3 independent experiments are shown in B–D. The statistical significance between the WT and each of the mutants was calculated using a one-way ANOVA with Tukey's multiple comparisons test using the area-under-the-curve values calculated for each sample; ns, no significant difference ($P > 0.05$). The data underlying this figure can be found in S2 Data. BMDM, bone marrow-derived macrophage; MOI, multiplicity of infection; PI, propidium iodide; T6SS, type VI secretion system.

either alone (Δ*tssM2*) or together with T6SS1 (Δ*hcp1*/Δ*tssM2*), completely abrogated the *V. coralliilyticus*-mediated cell death, whereas inactivation of T6SS1 (Δ*hcp1*) had either no effect (OCN008 and OCN014) or only a mild effect (BAA-450). These results support our hypothesis that *V. coralliilyticus* T6SS2 targets eukaryotes.

## T6SS1 and T6SS2 secrete diverse effector arsenals

Next, we performed comparative proteomics analyses to reveal the *V. coralliilyticus* T6SS secretomes and identify the effectors that mediate the antibacterial and anti-eukaryotic activities described above. Using mass spectrometry, we compared the proteins secreted by the wild-type *V. coralliilyticus* strains BAA-450, OCN008, and OCN014 with those secreted by their isogenic mutants in which either T6SS1 or T6SS2 have been inactivated (Δ*hcp1* or Δ*tssM2*, respectively).

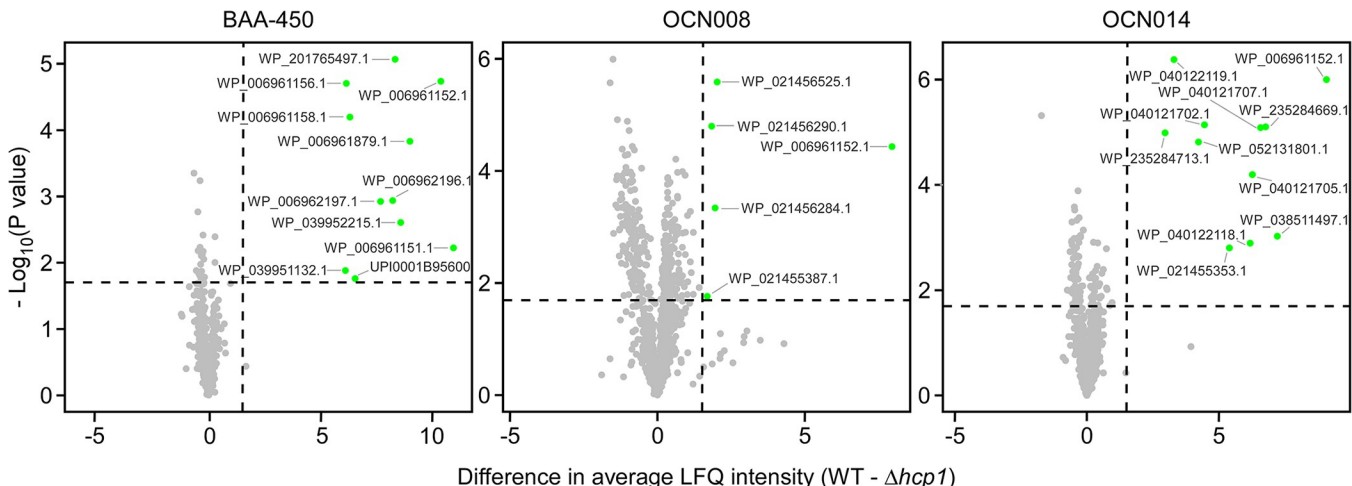

**Fig 5. *Vibrio coralliilyticus* T6SS1 effector repertoires.** Volcano plots summarizing the comparative proteomics of proteins identified in the media of the 3 indicated *V. coralliilyticus* strains with an active T6SS1 (WT, wild-type) or an inactive T6SS1 (Δ*hcp1*), using LFQ. The average LFQ signal intensity difference between the WT and Δ*hcp1* strains is plotted against the -Log$_{10}$ of Student's *t* test *P*-values (*n* = 3 biological replicates). Proteins that were significantly more abundant in the secretome of the WT strains (difference in average LFQ intensities > 1.6; *P*-value <0.02; with a minimum of 2 Razor unique peptides and Score >15) are denoted in green. The data underlying this figure can be found in S3 Data. LFQ, label-free quantification; T6SS, type VI secretion system.

**T6SS1 secretomes.** We identified 11, 6, and 11 proteins that were significantly enriched in the secretomes of wild-type strains BAA-450, OCN008, and OCN014, respectively, compared to their T6SS1⁻ (Δ*hcp1*) mutants (**Fig 5** and **Table 1** and **S3–S5 Dataset**). These include the secreted tube-spike structural components Hcp1 (which was deleted to inactivate T6SS1), VgrG1, and PAAR-like proteins. Most of the additional proteins are predicted antibacterial or anti-eukaryotic effectors, or proteins encoded next to them, including: (i) homologs of previously described T6SS effectors, with predicted toxic domains that target the peptidoglycan (e.g., WP_006961156.1 and WP_006961879.1); (ii) proteins containing MIX domains, which are markers for T6SS effectors [47], with predicted nuclease or pore-forming toxic domains (e.g., WP_039951132.1 and WP_201765497.1); and (iii) proteins that have yet to be described as related to T6SSs, which were identified only in the T6SS1 secretome of strain OCN008 (e.g., WP_021456284.1 and WP_021455387.1, which is a DEAD/DEAH box helicase). In accordance with our observation that the T6SS1 appears less active in strain OCN008 compared to the 2 other *V. coralliilyticus* strains under the assay conditions (**Fig 2A**), the comparative proteomics intensity difference for the putative OCN008 T6SS1 effectors was low (**Fig 5**), suggesting that the latter type of proteins detected only in the OCN008 T6SS1 secretome may be false positives. As previously reported for similar T6SSs in other vibrios [26,28,47,56], some of the identified proteins are encoded within the T6SS1 gene cluster, whereas others are encoded in auxiliary or orphan operons. Moreover, predicted antibacterial effectors are encoded next to putative immunity genes. Taken together, these results support our findings that *V. coralliilyticus* T6SS1 plays a role in interbacterial competitions using antibacterial effectors. Interestingly, in each *V. coralliilyticus* strain, we also identified a secreted MIX domain-containing effector that we previously showed or hypothesized targets eukaryotes rather than bacteria (e.g., WP_006962196.1) [22]. This finding suggests that T6SS1 also plays a role in interactions with eukaryotes, even though our experiments did not reveal significant T6SS1-mediated anti-eukaryotic effects.

**T6SS2 secretomes.** We identified 10, 9, and 6 proteins that were significantly enriched in the secretomes of wild-type strains BAA-450, OCN008, and OCN014, respectively, compared

**Table 1.** *Vibrio coralliilyticus* **T6SS1 secretomes identified by comparative proteomics.**

| Predicted role | Predicted activity or domain | BAA-450 | | OCN008 | | OCN014 | |
|---|---|---|---|---|---|---|---|
| | | Protein accession | Gene locus | Protein accession | Gene locus | Protein accession | Gene locus |
| **T6SS structural** | Hcp | WP_006961152.1 | VIC_RS16330 | WP_006961152.1 | G3U99_RS23805 | WP_006961152.1 | JV59_RS20030 |
| | VgrG | WP_006961151.1 | VIC_RS16325 | N/D | N/D | WP_040121702.1 | JV59_RS20025 |
| | PAAR-like (DUF4150) | WP_039952215.1 | VIC_RS19185 | N/D | N/D | WP_040122118.1 | JV59_RS22990 |
| | PAAR-like (DUF4150) | UPI0001B95600 (annotated as a pseudogene in RefSeq) | VIC_RS08805 | WP_021455353.1 | G3U99_RS15395 | WP_021455353.1 | JV59_RS10110 |
| **Antibacterial effector** | VP1390-like | WP_006961156.1 | VIC_RS16350 | WP_021456525.1 | G3U99_RS23785 | WP_040121705.1 | JV59_RS20045 |
| | Lysozyme-like | WP_006961879.1 | VIC_RS19190 | N/A | N/A | WP_040122119.1 | JV59_RS22995 |
| | MIX domain; TMs | WP_201765497.1 | VIC_RS12080 | N/D | N/D | WP_235284713.1 | JV59_RS24085 |
| | MIX domain; Pyocin_S; Colicin E9-like nuclease | WP_039951132.1 | VIC_RS01010 | N/A | N/A | N/A | N/A |
| | MIX domain; Colicin A-like pore-forming | N/A | N/A | N/A | N/A | WP_052131801.1 | JV59_RS24930 |
| | Unknown | N/A | N/A | WP_021456284.1 | G3U99_RS12660 | N/A | N/A |
| **Anti-eukaryotic effector** | MIX domain | WP_006962196.1 | VIC_RS20535 | N/A | N/A | N/A | N/A |
| | MIX domain | N/A | N/A | WP_021456290.1 | G3U99_RS26335 | WP_235284669.1 | JV59_RS27320 |
| **Effector accessory** | MIX domain-containing co-effector | WP_006961158.1 | VIC_RS16360 | N/D | N/D | WP_040121707.1 | JV59_RS20055 |
| | Encoded upstream of anti-eukaryotic MIX domain-containing effector | WP_006962197.1 | VIC_RS20540 | N/D | N/D | WP_038511497.1 | JV59_RS07245 |
| **Unknown** | DEAD/DEAH box helicase | N/D | N/D | WP_021455387.1 | G3U99_RS17670 | N/A | N/A |

N/A, no homolog is encoded in the genome; N/D, a homolog is encoded in the genome but not detected in the mass spectrometry analysis; TM, transmembrane helix (according to phobius).

to their T6SS2⁻ (Δ*tssM2*) mutants (**Fig 6** and **Table 2** and **S3–S5 Dataset**). These include the secreted tube-spike structural components Hcp2 and VgrG2. We predict that all the other identified, nonstructural proteins, which are encoded outside the T6SS2 gene cluster (**Fig E in S1 Text**), are novel anti-eukaryotic effectors (excluding the phage shock protein, WP_021456780.1, which is probably a phage protein and not a T6SS effector). In support of this prediction, none of these proteins is encoded next to a gene that could encode for a cognate immunity protein. Moreover, some are similar to previously described virulence toxins, such as WP_006960006.1 containing a predicted YopT-like cysteine protease domain (YopT is a type III secretion system virulence effector from *Yersinia* [57]). No putative effectors have a predicted signal peptide for the Sec or Tat secretion systems that could account for their secretion, according to SignalP 6.0 [58] analyses.

## T6SS2 effectors are novel anti-eukaryotic toxins

Since most of the T6SS1 effectors we identified are homologs of previously described effectors, we focused on the novel T6SS2 effectors for subsequent analyses. Altogether, the identified *V. coralliilyticus* T6SS2 effector repertoire comprises 9 putative novel effectors, which we named *Coralliilyticus* Virulence effector 1 to 9 (CoVe1-9): CoVe1, 2, 4, and 5 were identified in the secretomes of all 3 strains; CoVe6 was identified in the secretomes of BAA-450 and OCN008; CoVe3, 7, and 8 were identified only in the secretome of BAA-450, and CoVe9 was identified only in the secretome of OCN008 (**Table 2**).

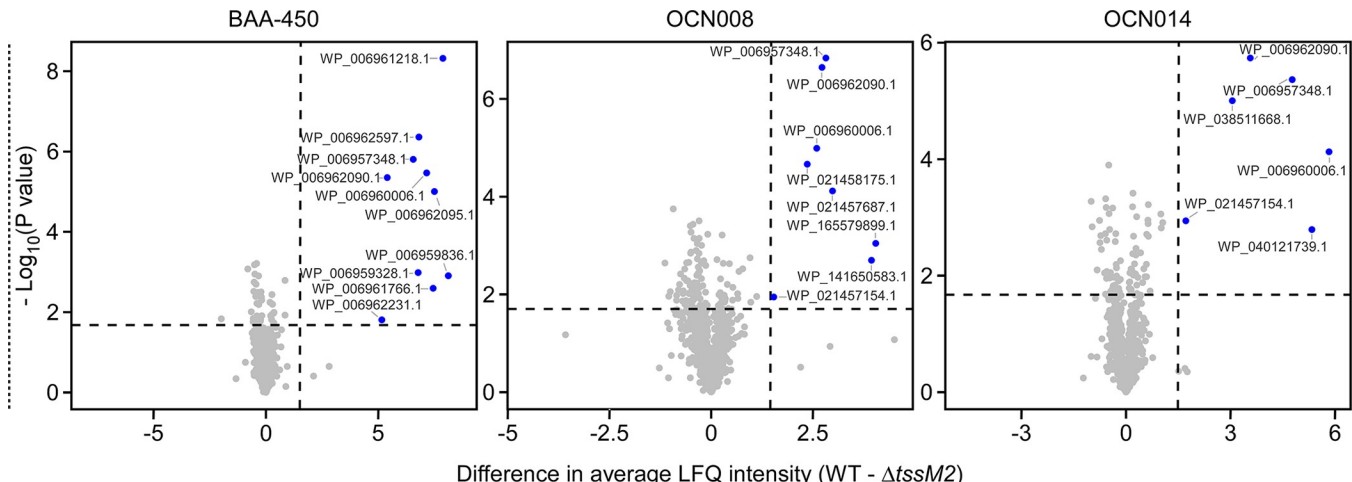

**Fig 6. *Vibrio coralliilyticus* T6SS2 effector repertoires.** Volcano plots summarizing the comparative proteomics of proteins identified in the media of the 3 indicated *V. coralliilyticus* strains with an active T6SS2 (WT, wild-type) or an inactive T6SS2 (Δ*tssM2*), using LFQ. The average LFQ signal intensity difference between the WT and Δ*tssM2* strains is plotted against the -Log$_{10}$ of Student's *t* test *P*-values ($n = 3$ biological replicates). Proteins that were significantly more abundant in the secretome of the WT strains (difference in average LFQ intensities >1.6; *P*-value <0.02; with a minimum of 2 Razor unique peptides and Score >15) are denoted in blue. The data underlying this figure can be found in S4 Data. LFQ, label-free quantification; T6SS, type VI secretion system.

Six of the 9 CoVes contain domains with predicted toxic activities (**Table 2**), including peptidase [57], ADP-ribosyltransferase [59], cytotoxic necrotizing factor (CNF)-like deamidase [60], and (p)ppGpp synthetase/hydrolase [61]. However, CoVe5, 8, and 9 sequence analyses did not reveal significant similarity to any previously investigated toxin, suggesting that they harbor novel toxic domains.

**Table 2. *Vibrio coralliilyticus* T6SS2 secretomes identified by comparative proteomics.**

| Predicted role | Predicted activity or domain | BAA-450 | | OCN008 | | OCN014 | |
|---|---|---|---|---|---|---|---|
| | | Protein accession | Gene locus | Protein accession | Gene locus | Protein accession | Gene locus |
| T6SS structural | Hcp | WP_006962090.1 | VIC_RS20130 | WP_006962090.1 | G3U99_RS13135 | WP_006962090.1 | JV59_RS07745 |
| | VgrG | WP_006962095.1 | VIC_RS20150 | WP_021458175.1 | G3U99_RS13115 | WP_038511668.1 | JV59_RS07720 |
| Anti-eukaryotic effector | CNF-likea (CoVe1) | WP_006957348.1 | VIC_RS01360 | WP_006957348.1 | G3U99_RS07885 | WP_006957348.1 | JV59_RS02730 |
| | (p)ppGpp synthetase / hydrolasea (CoVe2) | WP_006959328.1 | VIC_RS09310 | WP_021457154.1 | G3U99_RS15865 | WP_021457154.1 | JV59_RS10570 |
| | Cysteine peptidasea (CoVe3) | WP_006959836.1 | VIC_RS11210 | N/D | N/D | N/D | N/D |
| | peptidase_C58-like super familyb; TM (CoVe4) | WP_006960006.1 | VIC_RS11685 | WP_006960006.1 | G3U99_RS19905 | WP_006960006.1 | JV59_RS23695 |
| | Unknown (CoVe5) | WP_006961218.1 | VIC_RS16620 | WP_165579899.1 | G3U99_RS23535 | WP_040121739.1 | JV59_RS20315 |
| | ADP-ribosyltransferaseb (CoVe6) | WP_006961766.1 | VIC_RS18765 | WP_021457687.1 | G3U99_RS21080 | N/A | N/A |
| | Peptidase_26-likeb (CoVe7) | WP_006962231.1 | VIC_RS20705 | N/D | N/D | N/D | N/D |
| | TM (CoVe8) | WP_006962597.1 | VIC_RS22130 | N/D | N/D | N/D | N/D |
| | Unknown (CoVe9) | N/A | N/A | WP_141650583.1 | G3U99_RS19765 | N/A | N/A |
| Unknown | Phage shock protein PspA | N/D | N/D | WP_021456780.1 | G3U99_RS10060 | N/D | N/D |

N/A, no homolog is encoded in the genome; N/D, a homolog is encoded in the genome but not detected in the mass spectrometry analysis; TM, transmembrane helix (according to phobius).

a According to HHpred.

b According to NCBI CDD.

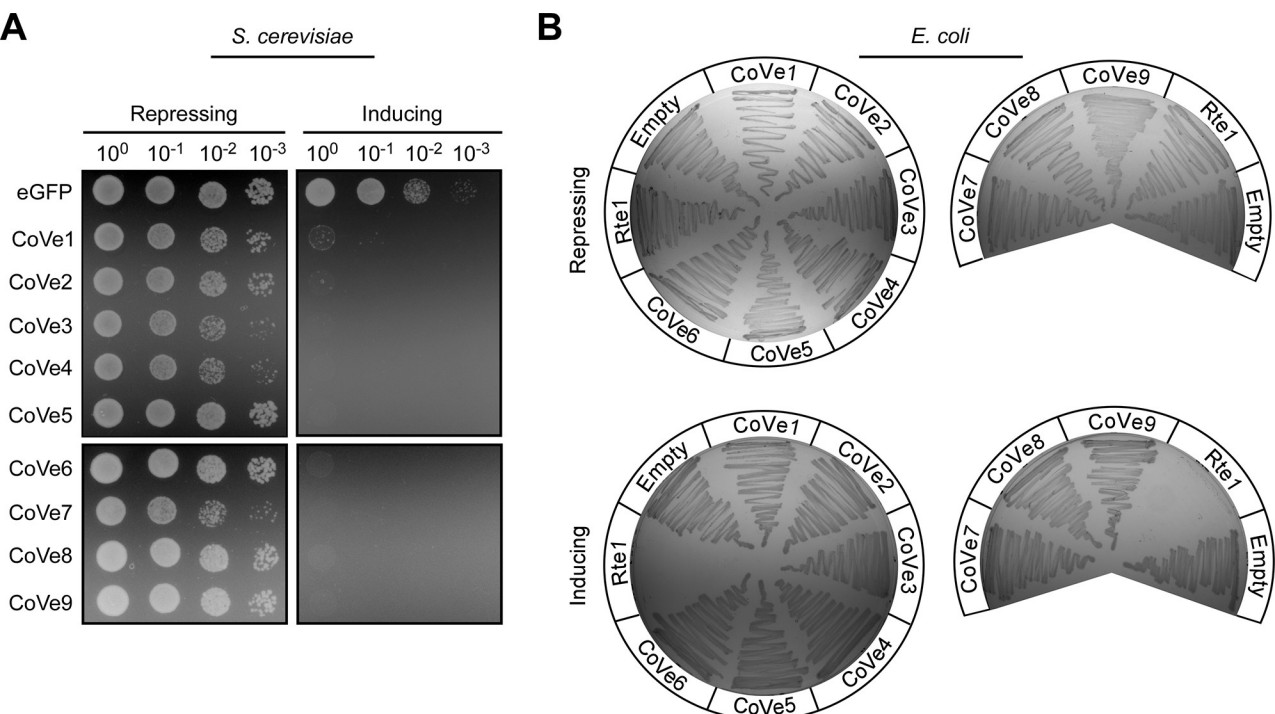

**Fig 7. *Vibrio coralliilyticus* T6SS2 effectors are toxic in eukaryotic cells. (A)** CoVes are toxic in yeast. Ten-fold serial dilutions of *S. cerevisiae* strains containing plasmids for the galactose-inducible expression of the indicated CoVes, or eGFP used as a negative control, were spotted on repressing (2% [wt/vol] glucose) or inducing (2% [wt/vol] galactose and 1% [wt/vol] raffinose) agar plates. eGFP, enhanced GFP. **(B)** CoVes are not toxic to bacteria. *E. coli* strains containing plasmids for the arabinose-inducible expression of the indicated, C-terminally FLAG-tagged CoVes, the *V. campbellii* antibacterial T6SS effector Rte1 used as a positive control, or an empty plasmid (Empty) were streaked onto repressing (0.4% [wt/vol] glucose) or inducing (0.001% [wt/vol] arabinose) agar plates. Results from a representative experiment out of at least 3 independent experiments are shown. T6SS, type VI secretion system.

We sought to investigate these putative effectors. First, we set out to further validate their T6SS2-dependent secretion using a standard secretion assay. To this end, we cloned the 9 putative effectors (CoVe1-8 from strain BAA-450 and CoVe9 from strain OCN008) into an arabinose-inducible expression plasmid, fused to a C-terminal FLAG tag, and monitored their secretion to the media from *V. coralliilyticus* strains. As shown in **Fig F in S1 Text**, T6SS2-dependent secretion of all CoVes, except CoVe3, was evident upon ectopic overexpression from a plasmid in their respective encoding *V. coralliilyticus* strain. Since CoVe3 T6SS2-dependent secretion was observed in the more sensitive comparative proteomics approach when endogenously expressed from the bacterial chromosome (**Fig 6**), it is possible that its overexpression from a plasmid hampered the secretion; alternatively, the C-terminal tag that we added to allow CoVe immunoblot detection may have interfered with CoVe3 secretion.

Next, we tested our hypothesis that these novel effectors target eukaryotes. In support of this hypothesis, we found that all 9 effectors are toxic when ectopically expressed from a galactose-inducible plasmid in a eukaryotic heterologous model organism, the yeast *Saccharomyces cerevisiae* [62,63] (**Fig 7A**). In contrast, these effectors were not toxic when expressed from an arabinose-inducible plasmid in *E. coli*, used as a surrogate model bacterium (**Figs 7B** and **G in S1 Text**). These results indicate that T6SS2 secretes an arsenal of novel effectors with anti-eukaryotic activities.

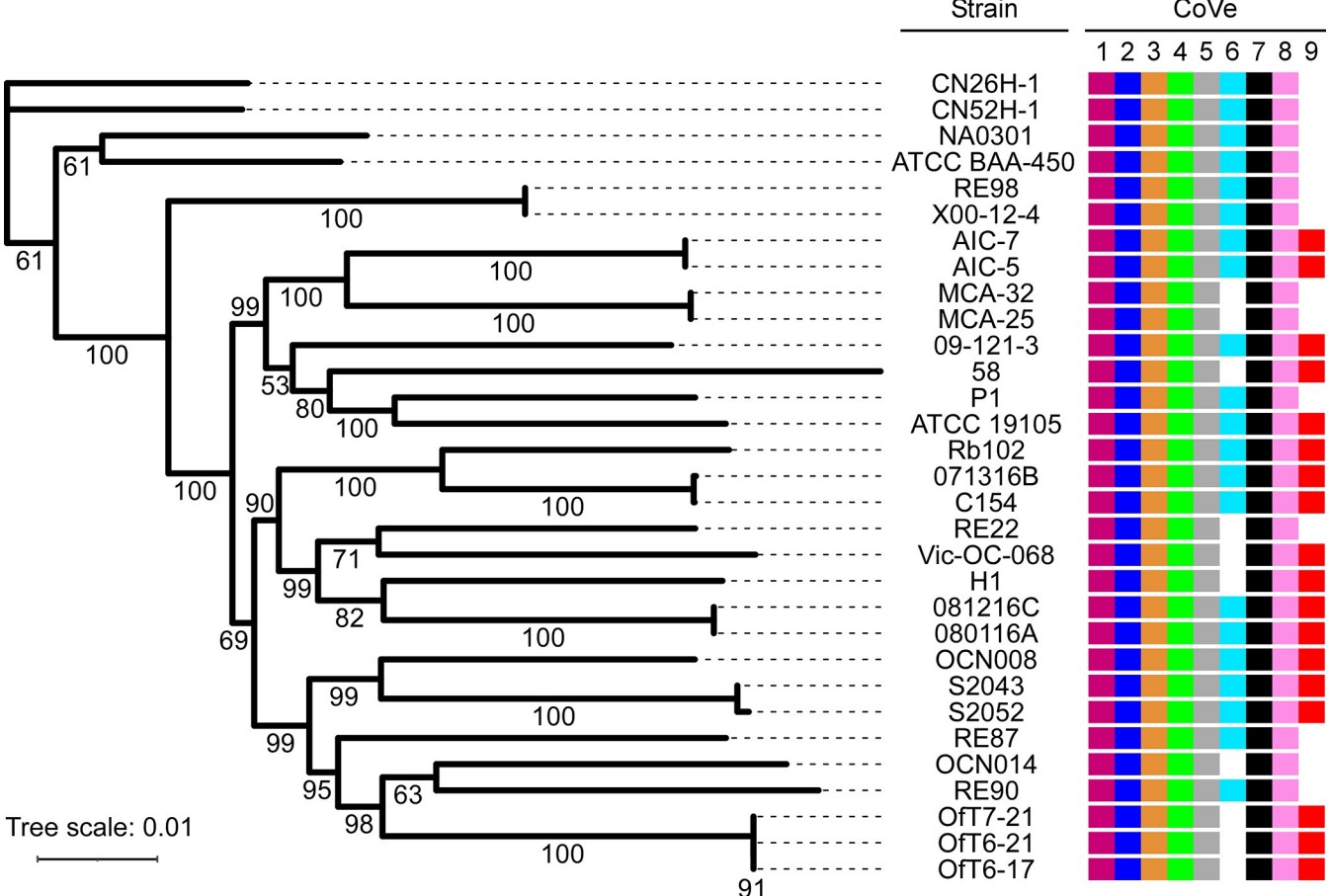

**Fig 8. The *Vibrio coralliilyticus* T6SS2 effector repertoire can be divided into core and accessory arsenals.** Distribution of T6SS2 CoVe1-9 effectors in RefSeq *V. coralliilyticus* genomes. The phylogenetic tree is based on a comparison of the codon sequences of 1,210 complete core genome proteins found in the indicated strains. The evolutionary history was inferred using the maximum likelihood method. Bootstrap values appear next to the corresponding branch as percent of 100 replicates. The data underlying this figure can be found in S5 Data. T6SS, type VI secretion system.

## T6SS2 effectors are differentially distributed in *Vibrio coralliilyticus* genomes

We and others previously showed that T6SS effector repertoires can be divided into core effectors present in all strains harboring the system and accessory effectors encoded only by a subset of strains [25,64,65]. Therefore, we sought to determine the distribution of CoVes in *V. coralliilyticus* genomes. Interestingly, 7 of the 9 CoVes are found in all available RefSeq *V. coralliilyticus* genomes (**Fig 8** and **S6 Dataset**). We propose that these 7 CoVes constitute the core effector repertoire of the *V. coralliilyticus* T6SS2. In contrast, 2 effectors, CoVe6 and CoVe9, are found only in a subset of strains, suggesting that they are part of the accessory T6SS2 effector repertoire. Interestingly, homologs of CoVe2 and CoVe8 are also found in all *V. coralliilyticus* genomes (**Fig E in S1 Text** and **S6 Dataset**). Even though these homologs were not identified in our comparative proteomics analyses, it is possible that they are also T6SS2 effectors.

## Discussion

*V. coralliilyticus* is a pathogen that inflicts devastating ecological and economic losses. Although environmental conditions, such as high temperatures, have been associated with

increased virulence and a pathogenic lifestyle, the virulence factors it uses remain poorly understood. Here, we systematically analyzed the T6SSs in the *V. coralliilyticus* pan-genome. We revealed 2 omnipresent T6SSs, T6SS1 and T6SS2, which are regulated by temperature and appear to contribute to *V. coralliilyticus* virulence. Whereas T6SS1 mediates antibacterial toxicity and thus possibly contributes to host colonization indirectly, T6SS2 secretes an array of novel anti-eukaryotic effectors and appears to play a direct role in virulence.

T6SS1 plays a role in interbacterial competition, possibly contributing to the elimination of commensal microbiota during host colonization. The T6SS1 effectors identified in our comparative proteomics analyses are homologs of effectors previously reported in similar T6SSs of other vibrios [22,28,47,48] or that have been predicted based on the presence of the MIX domain that defines a widespread class of polymorphic T6SS effectors [22,47]. Therefore, we did not investigate these effectors further in this work. Notably, we observed a mild effect of T6SS1 inactivation on the toxicity of the BAA-450 strain toward BMDMs, suggesting that T6SS1 may exert some effect also on eukaryotic cells. This could be mediated by the anti-eukaryotic MIX effector identified in the comparative proteomics analysis. However, we did not observe a similar effect with strains OCN008 and OCN014, even though they, too, secrete a predicted anti-eukaryotic MIX effector (**Table 1**).

T6SS2 secretes an array of anti-eukaryotic effectors and mediates toxicity during infection of a model host, *Artemia* nauplii, and during infection of macrophages. Because T6SS2 is induced at high temperatures, in correlation with the onset of *V. coralliilyticus* virulence, we propose that it plays a role in the colonization and toxicity towards its natural hosts, coral and shellfish larvae. In future work, we will investigate the contribution of T6SS2 to *V. coralliilyticus*'s virulence in these natural hosts, and we will also determine whether it targets the coral itself or its endosymbiotic dinoflagellates.

Notably, only a few anti-eukaryotic T6SS effectors are known [35,36]. Although we recently revealed anti-eukaryotic effectors in vibrios, belonging to the RIX effector class [42], the CoVes reported here do not belong to any known polymorphic effector class and appear to be new T6SS effectors that have not been previously described. Even though their activities and cellular targets remain to be investigated, 6 CoVes harbor putative catalytic domains that have previously been implicated in virulence. Future investigations will reveal their mechanism of action and targets inside eukaryotic cells, as well as their contribution to virulence.

Although we did not observe T6SS2 secretion in the poor GASW media in *V. coralliilyticus* OCN008, this system contributed to this strain's virulence during the infection of *Artemia* nauplii. *Artemia* nauplii were grown in poor media (i.e., Instant Ocean), yet the infection likely originated in the gastrointestinal tract of the animal, where nutrients may be more readily available. Host factors found in the animal gut may have also contributed to the activation of the T6SS during infection.

Notably, our results suggest that additional factors contribute to the virulence toward *Artemia*, since inactivation of T6SS2 did not completely abolish toxicity in this host. Interestingly, the results of the BMDM infection assays indicate that the *V. coralliilyticus* T6SS2 is also active at high temperatures of 37°C, which are infrequent in marine environments, and in low salinity media. It appears that under these conditions, the anti-eukaryotic toxicity of *V. coralliilyticus* strains is mediated predominantly by T6SS2, since its inactivation abrogated toxicity toward BMDMs. This observation suggests that *V. coralliilyticus* T6SS2 is functional under a wide temperature range and can overcome immune cells.

In conclusion, we find a new T6SS that specifically targets eukaryotes, and we identify its effector arsenal. Taken together with recent findings of vibrios that use T6SSs to target eukaryotes [22,40–45] and of widespread *Vibrio*-encoded anti-eukaryotic T6SS effectors [42], our results suggest that T6SSs should be considered as potential virulence factors. These findings

shed light on the molecular mechanisms that govern the connection between rising seawater temperatures and *V. coralliilyticus* virulence.

## Materials and methods

### Strains and media

For a complete list of strains used in this study, see **Table A in S1 Text**. *Escherichia coli* strain DH5α (λ-pir) was grown in 2xYT broth (1.6% [wt/vol] tryptone, 1% [wt/vol] yeast extract, and 0.5% [wt/vol] NaCl) or on lysogeny broth (LB) agar plates (1.5% [wt/vol]) at 37˚C. The media were supplemented with chloramphenicol (10 μg/ml) to maintain plasmids when needed. To repress expression from arabinose-inducible P*bad* promoters, 0.4% (wt/vol) D-glucose was added to the media. To induce expression from P*bad*, L-arabinose was added to the media at 0.001 or 0.1% (wt/vol), as indicated.

*V. coralliilyticus* strains ATCC BAA-450, OCN008, and OCN014, and their derivatives were grown in Marine Lysogeny broth (MLB; LB containing 3% [wt/vol] NaCl) or on GASW-Tris agar plates (20.8 [g/l] NaCl, 0.56 [g/l] KCl, 4.8 [g/l] $MgSO_4 \cdot 7H_2O$, 4 [g/l] $MgCl_2 \cdot 6H_2O$, 0.01 [g/l] $K_2HPO_4$, 0.001 [g/l] $FeSO_4 \cdot 7H_2O$, 2 [g/l] Instant Ocean sea salts, 6.33 [g/l] Tris base [$C_4H_{11}NO_3$], 4 [g/l] tryptone, 2 [g/l] yeast extract, 0.2% [vol/vol] glycerol, and 1.5% [wt/vol] agar; pH was adjusted to 8.3 with HCl) at 30˚C. For colony selection after plasmid conjugation (see below), *V. coralliilyticus* was grown on TCBS agar (Millipore, #86348) plates. L-arabinose (0.01% [wt/vol]) was added to the media to induce expression from P*bad*.

*Vibrio natriegens* ATCC 14048 were grown on Marine Minimal Media (MMM) agar plates (2% [wt/vol] NaCl, 0.4% [wt/vol] galactose, 5 mM $MgSO_4$, 7 mM $K_2SO_4$, 77 mM $K_2HPO_4$, 35 mM $KH_2PO_4$, 2 mM $NH_4Cl$, and 1.5% [wt/vol] agar) at 30˚C. The media were supplemented with chloramphenicol (10 μg/ml) to select for or maintain plasmids when necessary.

*Saccharomyces cerevisiae* were grown in Yeast Extract–Peptone–Dextrose broth (YPD; 1% [wt/vol] yeast extract, 2% [wt/vol] peptone, and 2% [wt/vol] glucose) or on YPD agar plates (2% [wt/vol]) at 30˚C. Yeast containing plasmids that provide prototrophy to leucine were grown in Synthetic Dropout media (SD; 6.7 [g/l] yeast nitrogen base without amino acids, 1.4 [g/l] yeast synthetic dropout medium supplement (Sigma)) supplemented with histidine (2 [ml/l] from a 1% [wt/vol] stock solution), tryptophan (2 [ml/l] from a 1% [wt/vol] stock solution), uracil (10 [ml/l] from a 0.2% [wt/vol] stock solution), and glucose (4% [wt/vol]). For galactose-inducible expression from a plasmid, cells were grown in SD media or on SD agar plates supplemented with galactose (2% [wt/vol]) and raffinose (1% [wt/vol]).

### Plasmid construction

For a complete list of plasmids used in this study, see **Table B in S1 Text**. For a complete list of primers used in this study, see **Table C in S1 Text**. To enable strong, arabinose-inducible protein expression in *V. coralliilyticus*, we constructed the plasmid pKara1. To this end, we amplified the region between the *araC* cassette and *rrnB* T1 terminator, including a C-terminally FLAG-tagged sfGFP gene, from the plasmid psfGFP [48], and introduced it 220 bp upstream of the gene encoding the fluorescent protein DsRed in pVSV208 [66], using the Gibson assembly method.

For expression in bacteria, the coding sequences (CDS) of the indicated genes of interest were amplified by PCR from the respective genomic DNA of the encoding bacterium. Next, amplicons were inserted into the multiple cloning site (MCS) of pBAD33.1$^F$, or in place of the sfGFP gene within pKara1, using the Gibson assembly method [67], in-frame with the C-terminal FLAG tag. Plasmids were introduced into *E. coli* DH5α (λ-pir) by electroporation and

into vibrios via conjugation. Transconjugants were selected on TCBS agar (Millipore) plates supplemented with chloramphenicol.

For galactose-inducible expression in yeast, genes were inserted into the MCS of the shuttle vector pGML10 (Riken) using the Gibson assembly method, in-frame with a C-terminal Myc tag. Yeast transformations were performed using the lithium acetate method, as described previously [68].

### Construction of deletion strains

To delete genes in *V. coralliilyticus* BAA-450, OCN008, and OCN014, 1 kb sequences upstream and downstream of each gene to be deleted were cloned together into the MCS of pDM4, a Cm$^R$OriR6K suicide plasmid. The pDM4 constructs were transformed into *E. coli* DH5α (λ-pir) by electroporation and then conjugated into *V. coralliilyticus* strains. Transconjugants were selected on TCBS agar plates supplemented with chloramphenicol and then counter-selected on agar plates containing 15% (wt/vol) sucrose for loss of the *sacB*-containing plasmid. Deletions were confirmed by PCR.

### *Vibrio* protein secretion assays

Secretion assays were performed as previously reported [24], with minor modifications. *V. coralliilyticus* strains were grown for 16 h in MLB supplemented with antibiotics to maintain plasmids when necessary. Bacterial cultures were diluted 4-fold in fresh media and incubated for 2 additional hours at 28°C. Then, the cultures were normalized to an optical density at 600 nm (OD$_{600}$) of 0.18 in 5 ml of MLB or GASW media, as indicated. When protein expression from an arabinose-inducible plasmid was required, the media were supplemented with chloramphenicol and 0.01% (wt/vol) L-arabinose. The cultures were then incubated with continuous shaking (220 rpm) at 19°C, 23°C, 28°C, or 31°C, as indicated, for 4 h. For expression fractions, 0.5 OD$_{600}$ units were harvested, and cell pellets were resuspended in 30 μl of 2× Tris-glycine SDS sample buffer (Novex, Life Sciences) with 5% (vol/vol) β-mercaptoethanol. For secretion fractions, supernatant volumes equivalent to 5 OD$_{600}$ units were filtered (0.22 μm), and proteins were precipitated using the deoxycholate and trichloroacetic acid method [69]. The precipitated proteins were washed twice with cold acetone and air-dried before being resuspended in 20 μl of 100 mM Tris-Cl (pH = 8.0) and 20 μl of 2× Tris-glycine SDS sample buffer containing 5% (vol/vol) β-mercaptoethanol. Protein samples were incubated at 95°C for 10 min before being resolved on TGX Stain-free gels (Bio-Rad). The proteins were transferred onto 0.2 μm nitrocellulose membranes using Trans-Blot Turbo Transfer (Bio-Rad), following the manufacturer's protocol. Membranes were then immunoblotted with custom-made α-Hcp2 (GenScript; polyclonal antibodies raised in rabbits against the peptides CGEGGKIEKG-PEVGF or CVMTKPNREGSGADP; the latter was used only in the experiment shown in Fig 2A), Custom-made polyclonal α-VgrG1 [50], monoclonal α-FLAG (Sigma-Aldrich, F1804), or Direct-Blot HRP anti-*E. coli* RNA polymerase sigma 70 (mouse mAb #663205; Bio-Legend; referred to as α-RNAP) antibodies at a dilution of 1:1,000. Protein signals were detected using enhanced chemiluminescence (ECL) reagents with a Fusion FX6 imaging system (Vilber Lourmat).

### Mass spectrometry analyses

Sample preparations for mass spectrometry were performed as described in the "*Vibrio* protein secretion assays" section. After the acetone wash step, samples were shipped to the Smoler Proteomics Center at the Technion, Israel, for analysis. Precipitated proteins were washed twice in 80% (vol/vol) cold acetone. The protein pellets were dissolved in 8.5 M Urea, 400 mM

ammonium bicarbonate, and 10 mM DTT. Protein concentrations were estimated using the Bradford assay. The proteins were reduced at 60°C for 30 min and then modified with 35.2 mM iodoacetamide in 100 mM ammonium bicarbonate for 30 min at room temperature in the dark. The proteins were digested overnight at 37°C in 1.5 M urea and 66 mM ammonium bicarbonate with modified trypsin (Promega) at a 1:50 (M/M) enzyme-to-substrate ratio. An additional trypsinization step was performed for 4 h. The resulting tryptic peptides were analyzed by LC-MS/MS using Q Exactive HF mass spectrometer (Thermo) fitted with a capillary HPLC (Evosep). The peptides were loaded onto a 15 cm ID 150 1.9-micron (Batch no. E1121-3-24) column of Evosep. The peptides were eluted with the built-in Xcalibur 15 SPD (88 min) method. Mass spectrometry was performed in a positive mode using repetitively full MS scan (m/z 350 to 1,200) followed by High Energy Collision Dissociation (HCD) of the 20 most dominant ions selected from the full MS scan. A dynamic exclusion list was enabled with exclusion duration of 20 s.

The mass spectrometry data were analyzed with the MaxQuant software 2.1.1.0 (www. maxquant.org) using the Andromeda search engine [70] against the relevant *V. coralliilyticus* strains from the Uniprot database, with a mass tolerance of 4.5 ppm for the precursor masses and 4.5 ppm for the fragment ions. Peptide- and protein-level false discovery rates (FDRs) were filtered to 1% using the target-decoy strategy. The protein table was filtered to eliminate identities from the reverse database and common contaminants. The data were quantified by label-free analysis using the same software, based on extracted ion currents (XICs) of peptides, enabling quantitation from each LC/MS run for each peptide identified in any of the experiments. Statistical analyses of the identification and quantization results were done using the Perseus 1.6.7.0 software [71]. The mass spectrometry proteomics data have been deposited in the ProteomeXchange Consortium via PRIDE [72].

## Bacterial competition assays

Bacterial competition assays were performed as previously described [24], with minor modifications. Attacker and prey strains were grown for 16 h in appropriate media. In the morning, *V. coralliilyticus* attacker strains were diluted 1:10 into fresh media and incubated for an additional hour at 28°C. Attacker and prey cultures were then normalized to an $OD_{600}$ of 0.5 and mixed at a 4:1 (attacker:prey) ratio in triplicate. Next, the mixtures were spotted (25 μl) on MLB agar competition plates and incubated at 28°C for 4 h. The colony-forming units (CFU) of the prey strains at t = 0 h were determined by plating tenfold serial dilutions on selective media plates. After 4 h of co-incubation on competition plates, the bacteria were harvested, and the CFUs of the surviving prey strains were determined as described above. Prey strains contained a pBAD33.1 [73] (*V. natriegens* and *V. alginolyticus*) or pVSV208 [66] (*V. campbellii*) plasmid to allow selective growth on plates containing chloramphenicol.

## *Vibrio coralliilyticus* growth assays

Triplicates of *V. coralliilyticus* cultures grown for 16 h were normalized to $OD_{600}$ = 0.01 in MLB and transferred to a 96-well plate (200 μl per well). The 96-well plate was incubated in a microplate reader (BioTek SYNERGY H1) at 28°C with continuous shaking (205 cpm). Growth was measured as $OD_{600}$ in 10-min intervals.

## *Artemia* infection assays

*Artemia* infection assays were performed as previously reported [41], with minor modifications. *Artemia salina* eggs (Artemio Pur; JBL) were incubated in deionized distilled water containing chloramphenicol (10 μg/ml), kanamycin (100 μg/ml), and ampicillin (100 μg/ml) at

28˚C with continuous rotation for an hour. The eggs were washed 4 times with Instant Ocean solution (3.3% [wt/vol]; Aquarium Systems) and then incubated for 24 h with continuous rotation at 28˚C. Hatched *Artemia* nauplii were transferred into sterile 48-well plates (2 nauplii per well in 400 µl Instant Ocean). Approximately $5 \times 10^7$ bacteria were added to each well, and the plates were incubated at 28˚C under 12-h light and dark cycles. *Artemia* survival was determined at the indicated time points postinfection. An *Artemia* nauplius that did not move for 10 s was defined as nonviable. Each bacterial strain was added to 8 wells (16 nauplii). Survival results are provided as grouped data from 4 independent experiments. Percent survival was calculated as surviving subjects out of the subjects at risk for each time point.

## BMDM infection assays

Bone marrow cells from 6- to 8-week-old mice were isolated, and BMDMs were obtained after a 7-day differentiation, as previously described [74]. *V. coralliilyticus* strains were grown for 16 h in MLB. In the morning, bacterial cultures were diluted tenfold into fresh media and incubated for an additional hour at 28˚C. Approximately $3.5 \times 10^4$ BMDMs were seeded into 96-well plates in triplicates in 1% (vol/vol) FBS and penicillin-streptomycin-free DMEM media and then infected with the indicated *V. coralliilyticus* strains at a multiplicity of infection (MOI) ~ 4. Plates were centrifuged for 5 min at $400 \times g$. Propidium iodide (PI; 1 µg/ml) was added to the medium 30 min prior to infection, and its uptake kinetics were assessed every 15 min using real-time microscopy (Incucyte SX5) during incubation at 37˚C. The data were analyzed using the Incucyte SX5 analysis software and exported to Graphpad PRISM. Normalization was performed according to the maximal PI-positive object count to calculate the percentage of dead cells [74].

## Yeast toxicity assays

Toxicity assays in yeast were performed as previously described [68]. Briefly, yeast cells were cultured for 16 h in SD media supplemented with 4% glucose (wt/vol). Yeast cultures were washed twice with sterile deionized distilled water and normalized to an $OD_{600}$ of 1.0 in sterile deionized water. Then, 10-fold serial dilutions were spotted onto SD agar plates containing 4% (wt/vol) glucose (repressing plates) or 2% (wt/vol) galactose and 1% (wt/vol) raffinose (inducing plates). The plates were incubated at 28˚C for 2 days.

## Protein expression in *E. coli*

Overnight-grown bacterial cultures of *E. coli* DH5α (λ-pir) strains carrying pBAD33.1 arabinose-inducible expression plasmids were grown in 2xYT broth supplemented with chloramphenicol. Bacterial cultures were normalized to an $OD_{600} = 0.5$ in 3 ml fresh 2xYT with chloramphenicol and incubated with continuous shaking (220 rpm) at 37˚C for 2 h. Then, L-arabinose was added to a final concentration of 0.1% (wt/vol) to induce protein expression, and the cultures were incubated for 2 additional hours. Cells equivalent to 0.5 $OD_{600}$ units were harvested, and their pellets were resuspended in 50 µl of 2× Tris-glycine SDS sample buffer (Novex, Life Sciences) supplemented with 5% (vol/vol) β-mercaptoethanol. Subsequently, the samples were boiled at 95˚C for 10 min and resolved on a TGX Stain-free gel (Bio-Rad) for SDS-PAGE analysis. The proteins were transferred onto nitrocellulose membranes, which were then immunoblotted with α-FLAG (Sigma-Aldrich, F1804) antibodies at a 1:1,000 dilution. Finally, protein signals were detected using ECL in a Fusion FX6 imaging system (Vilber Lourmat). The loading control for total protein lysates was visualized as the fluorescence of activated trihalo compounds found in the gel.

## *E. coli* toxicity assays

To determine the toxicity of *V. coralliilyticus* proteins in bacteria, *E. coli* DH5α (λ-pir) strains carrying pBAD33.1 arabinose-inducible expression plasmids were streaked onto LB agar plates supplemented with chloramphenicol and either 0.4% (wt/vol) glucose (repressing plates) or 0.001% (wt/vol) L-arabinose (inducing plates). Plates were incubated for 16 h at 37°C.

## Identifying T6SS gene clusters in *Vibrio coralliilyticus*

A local database containing the RefSeq bacterial nucleotide and protein sequences was generated (last updated on August 21, 2023). *V. coralliilyticus* genomes under NCBI Taxonomy ID = 190893 were retrieved from the local database, and OrthoANI [75] was performed as described previously [76]. The *V. coralliilyticus* strain SCSIO 43001 genome (assembly accession GCF_024449095.1) was removed from the data set because it showed OrthoANI values <95%. The *V. coralliilyticus* strain RE22 (assembly accession GCF_001297935.1) was removed because an updated version of strain RE22 was found (assembly accession GCF_003391375.1).

The presence of T6SS gene clusters in *V. coralliilyticus* genomes was determined by following a two-step procedure described previously [56]. Briefly, in the first step, BLASTN was employed to align *V. coralliilyticus* nucleotide sequences against the nucleotide sequences of representative T6SS clusters (**Fig 1** and **S2 Dataset**). The best alignments for each nucleotide accession number were saved. In the second step, a 2D matrix was generated for each T6SS gene cluster. The matrices were filled in with the percent identity values based on the positions of the alignments from the first step. The overall coverage was calculated for each T6SS gene cluster in each genome. *V. coralliilyticus* genomes with at least 70% overall coverage of a T6SS gene cluster were regarded as containing that T6SS gene cluster (**S2 Dataset**). Comparative gene cluster analyses to determine the synteny of T6SS1 and T6SS2 were performed using CLINKER [77]. GenBank files used for CLINKER analyses were retrieved from NCBI.

## Identifying effector homologs in *Vibrio coralliilyticus* genomes

BLASTP was employed to identify homologs of the T6SS2 effectors in *V. coralliilyticus* genomes, as described previously [25]. The amino acid sequences of new CoVes from strains BAA-450 (WP_006957348.1, WP_006959328.1, WP_006959836.1, WP_006960006.1, WP_006961218.1, WP_006961766.1, WP_006962231.1, and WP_006962597.1) and OCN008 (WP_141650583.1) were used as queries. The E-value threshold was set to $10^{-12}$, and the coverage was set to 70% based on the length of the query sequences.

## Constructing a phylogenetic tree

The 1,445 core gene BUSCO definitions from vibrionales_odb10 were searched against the protein sequences for the 31 *V. coralliilyticus* strains using BUSCO version 4.0.5 [78]. Of these, 1,210 were found to be complete and non-duplicated in all 31 strains. Multiple sequence alignments were generated from the protein sequences homologous to each common core BUSCO using MUSCLE ver. 3.8.31 [79] with parameters: "-diags -sv -distance1 kbit20_3", which were then converted to nucleotide space by substituting each amino acid with the corresponding codon sequence from the associated coding sequences. The codon alignments were concatenated together, and a best scoring maximum likelihood tree was drawn using RAxML version 8.2.12 [80] along with bootstrapping from 100 bootstrap replicates (parameters: -m GTRGAMMA -N 100). The resulting tree was visualized using iTOL (https://itol.embl.de) [81].

## Supporting information

**S1 Text Fig A. T6SS1 clusters share a similar synteny.** Comparison of T6SS1 clusters in the 31 *V. coralliilyticus* strains analyzed in this study. Colors denote homology between the encoded protein sequences. The strain name and RefSeq accession are denoted. Dashed, black vertical lines denote borders between separate contigs. **Fig B. T6SS2 clusters share a similar synteny.** Comparison of T6SS2 clusters in the 31 *V. coralliilyticus* strains analyzed in this study. Colors denote homology between the encoded protein sequences. The strain name and RefSeq accession are denoted. Dashed, black vertical lines denote borders between separate contigs. **Fig C. Deletion of *hcp1* or *tssM2* inactivates T6SS1 or T6SS2, respectively. (A)** Expression (cells) and secretion (media) of VgrG1 and Hcp2 from the indicated *V. coralliilyticus* strains grown for 4 h at 28˚C in rich media (MLB). RNA polymerase sigma 70 (RNAp) was used as a loading and lysis control. Asterisks denote expected protein sizes. WT, wild-type. **(B)** The growth of the indicated *V. coralliilyticus* strains in MLB at 28˚C measured as absorbance at 600 nm ($OD_{600}$). Data are shown as the mean ± SD; $n$ = 3. Results from a representative experiment out of at least 3 independent experiments are shown. The data underlying panel B of this figure can be found in S6 Data. **Fig D. *Vibrio coralliilyticus* T6SS1 mediates interbacterial competition. (A, B)** Viability counts (colony forming units; CFU) of *V. alginolyticus* 12G01 and *V. campbellii* ATCC 25920 prey strains before (0 h) and after (4 h) co-incubation with the indicated *V. coralliilyticus* OCN008 attacker strains on MLB plates at 28˚C. The statistical significance between samples at the 4 h time point was calculated using an unpaired, two-tailed Student's $t$ test; WT, wild-type; DL, the assay's detection limit. Data are shown as the mean ± SD; $n$ = 3. The data shown are a representative experiment out of 3 independent experiments. The data underlying this figure can be found in S7 Data. **Fig E. Non-structural proteins secreted by *Vibrio coralliilyticus* T6SS2 are encoded by orphan genes.** Genomic neighborhoods of genes encoding representative T6SS effector proteins (CoVes) and their homologs (colored arrows). The strain names, the GenBank accession numbers, and protein accessions are denoted. Genes are denoted by arrows indicating the predicted direction of transcription. Gray rectangles denote regions of amino acid sequence homology; identity percentages are indicated. **Fig F. CoVes are secreted in a T6SS2-dependent manner.** Expression (cells) and secretion (media) of C-terminally FLAG-tagged CoVes expressed from arabinose-inducible plasmids in *V. coralliilyticus* strains, either wild-type (WT) or T6SS2⁻ (Δ*tssM2*). CoVe1-8 were monitored in *V. coralliilyticus* BAA-450 and CoVe9 was monitored in *V. coralliilyticus* OCN008. *V. coralliilyticus* strains were grown in MLB supplemented with chloramphenicol and 0.01% (wt/vol) L-arabinose for 4 h at 28˚C. Loading control (LC) is shown for total protein lysate. Results from a representative experiment out of at least 2 independent experiments are shown. **Fig G. CoVes are expressed in *E. coli*.** Expression of C-terminally FLAG-tagged CoVes from arabinose-inducible plasmids in *E. coli* strain DH5α (λ-pir). Loading control (LC) is shown for total protein lysate. Results from a representative experiment out of at least 2 independent experiments are shown. **Table A. Bacteria and yeast strains used in this study. Table B. Plasmids used in this study. Table C. Primers used in this study.** (DOCX)

**S1 Dataset. OrthoANI analysis of *V. coralliilyticus* strains used in this study.** (XLSX)

**S2 Dataset. Analysis of *V. coralliilyticus* T6SS gene clusters.** (XLSX)

**S3 Dataset. Mass spectrometry results for *V. coralliilyticus* BAA-450 samples.** (XLSX)

**S4 Dataset. Mass spectrometry results for *V*. *coralliilyticus* OCN008 samples.**
(XLSX)

**S5 Dataset. Mass spectrometry results for *V*. *coralliilyticus* OCN014 samples.**
(XLSX)

**S6 Dataset. CoVe distribution in RefSeq *V*. *coralliilyticus* genomes.**
(XLSX)

**S1 Data. Numerical values for Fig 3.**
(XLSX)

**S2 Data. Numerical values for Fig 4.**
(XLSX)

**S3 Data. Numerical values for Fig 5.**
(XLSX)

**S4 Data. Numerical values for Fig 6.**
(XLSX)

**S5 Data. Tree data for Fig 8.**
(TXT)

**S6 Data. Numerical values for panel B of Fig C in S1 Text.**
(XLSX)

**S7 Data. Numerical values for Fig D in S1 Text.**
(XLSX)

**S1 Raw Images. Uncropped and minimally adjusted images supporting all blot results included in the article.**
(PDF)

## Acknowledgments

We thank members of the Salomon, van Kessel, and Ushijima groups for valuable discussions, and Katarzyna Kanarek for preparing the pKara1 plasmid. We also thank the Smoler Proteomics Center at the Technion for performing and analyzing the mass spectrometry data.

## Author Contributions

**Conceptualization:** Blake Ushijima, Julia C. van Kessel, Dor Salomon.

**Formal analysis:** Shir Mass, Hadar Cohen, Ram Podicheti, Douglas B. Rusch, Eran Bosis, Dor Salomon.

**Funding acquisition:** Motti Gerlic, Blake Ushijima, Julia C. van Kessel, Eran Bosis, Dor Salomon.

**Investigation:** Shir Mass, Hadar Cohen, Eran Bosis.

**Methodology:** Shir Mass, Hadar Cohen, Eran Bosis.

**Resources:** Ram Podicheti, Douglas B. Rusch, Motti Gerlic, Blake Ushijima, Julia C. van Kessel.

**Supervision:** Dor Salomon.

**Writing – original draft:** Shir Mass, Dor Salomon.

**Writing – review & editing:** Hadar Cohen, Ram Podicheti, Douglas B. Rusch, Motti Gerlic, Blake Ushijima, Julia C. van Kessel, Eran Bosis.

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
