## [Editor Report · Decision Letter 0]

2 Apr 2024

Dear Dr Salomon, 

Thank you for submitting your manuscript entitled "A T6SS in the coral pathogen Vibrio coralliilyticus secretes an arsenal of anti-eukaryotic effectors and contributes to virulence" for consideration as a Research Article by PLOS Biology. I would like to apologize for the long time it took me to send you an initial decision. 

Your manuscript has now been evaluated by the PLOS Biology editorial staff, as well as by an academic editor with relevant expertise, and I am writing to let you know that we would like to send your submission out for external peer review.

Once your full submission is complete, your paper will undergo a series of checks in preparation for peer review. After your manuscript has passed the checks it will be sent out for review. To provide the metadata for your submission, please Login to Editorial Manager (https://www.editorialmanager.com/pbiology) within two working days, i.e. by Apr 04 2024 11:59PM.

Kind regards,

Melissa

Melissa Vazquez-Hernandez, Ph.D.

Associate Editor

PLOS Biology

---

## [Decision Letter · Decision Letter 1]

21 May 2024

Dear Dr Salomon,

Thank you for your patience while your manuscript "A T6SS in the coral pathogen Vibrio coralliilyticus secretes an arsenal of anti-eukaryotic effectors and contributes to virulence" went through peer-review at PLOS Biology. Your manuscript has now been evaluated by the PLOS Biology editors, an Academic Editor with relevant expertise, and by three independent reviewers, with reviewers Xiaoxue Wang (R1) and Frederique Le Roux (R2) identifying themselves. I would like to apologize for the delay in giving you a decision. 

In light of the reviews, which you will find at the end of this email, we are pleased to offer you the opportunity to address the comments from the reviewers in a revision that we anticipate should not take you very long. As you will see in the reports, all reviewers are positive about the relevance of the work. Reviewer #1 and #2 suggest to evaluate the pathogenicity of effector mutants, and Reviewer also #1 suggests to perform the competition experiment in a more representative environment. However, while we believe the experimental requests would strengthen the conclusions, they are not required for publication in our view. We leave it to your discretion if you want to include these or not. All other reviewer concerns and suggestions should be addressed for further consideration.

We will then assess your revised manuscript and your response to the reviewers' comments with our Academic Editor aiming to avoid further rounds of peer-review, although might need to consult with the reviewers, depending on the nature of the revisions.

**IMPORTANT - SUBMITTING YOUR REVISION**

*Resubmission Checklist*

*Published Peer Review*

*PLOS Data Policy*

*Blot and Gel Data Policy*

Sincerely,

Melissa

Melissa Vazquez Hernandez, Ph.D.

Associate Editor

PLOS Biology

REVIEWERS' COMMENTS:

Reviewer #1: 

The manuscript by Shir Mass et al., titled "A T6SS in the coral pathogen Vibrio coralliilyticus secretes an arsenal 2 of anti-eukaryotic effectors and contributes to virulence," analyzed the T6SS in Vibrio coralliilyticus and identified two omnipresent T6SS systems, T6SS1 and T6SS2. T6SS1 is involved in interbacterial competition, while T6SS2 mediates anti-eukaryotic toxicity and contributes to mortality during infection of Artemia salina. Through comparative proteomics, they identified the effectors of T6SS1 and T6SS2 in three Vcor strains, including nine novel effectors secreted by T6SS2. The results of this study are interesting and informative. Overall, it is a well-organized and well-written manuscript with rigorous logic.

There is a bit room for improvement and I have only two minor comments: 

1. The only downside is the lack of coral infection assay mediated by T6SS2 effectors.

On line 122, the authors performed comparative proteomics analyses to reveal the Vcor T6SS secretomes and identify the effectors. If possible, the mutant strains by deleting genes encoding effectors like toxic proteins could be constructed and used to test the pathogenicity for Artemia salina.

2. The authors competed Vcor strains against a sensitive a V. natriegens prey strain on rich media. It will be better if the selected vibrio strain is more representative or isolated in the natural environment, such as coral tissue, gastric cavity of shellfish larvae.

BTW, on line 164, "a sensitive a V. natriegens" into "a sensitive V. natriegens".

Reviewer #2: 

This study presents compelling evidence that two Type VI Secretion Systems (T6SS) are conserved in Vibrio coralliilyticus, a pathogen affecting corals and oysters. Both T6SS systems are activated at ecologically relevant temperatures (28°C), yet they differ in their secretion profiles: T6SS1 targets prokaryotes, while T6SS2 secretes anti-eukaryotic effectors. Notably, T6SS2 induces mortality in artemia, serving as a model system for pathogenicity assessment. Through comparative proteomics, the authors identified nine novel putative toxins that specifically target eukaryotic cells, shedding light on the diversity of T6SS effectors.

Overall, the data presented are convincing and innovative, supported by a well-designed experimental approach. This study underscores the vast potential for uncovering new insights into T6SS dynamics using non-human Vibrio pathogens.

I only have minor comments

Introduction

The assertion regarding the spread of vibrios to new regions correlating with rising ocean temperatures might benefit from rephrasing to avoid potential misinterpretation. Indeed I disagree with this statement. Vibrios encompass a wide range of species inhabiting diverse environments, including psychrophiles. There is a tendency for publications to conflate human pathogens like V. cholerae and V. parahaemolyticus with the broader category of vibrios, because research historically focused heavily on V. cholerae.

Additionally, please use "V. coralliilyticus" instead of "Vcor" everywhere

References to Rubio et al. (PNAS, 2019) and works from the Blokesch lab, particularly Drebes Dörr et al. (Environmental Microbiology, 2020), should be included for comprehensive coverage.

Results: 

Consider incorporating a figure depicting the core genome phylogeny of the 31 V. coralliilyticus strains and the synteny of T6SS1 and T6SS2 to facilitate data interpretation.

Clarify the term "Marine LB" (Line 134) for improved understanding.

Regarding the comparison of T6SS activity between rich and poor media at 28°C (Line 142), adjust the statement to reflect the relative protein abundance differences more accurately (clearly less proteins in poor media).

For Figures 4B, C, D, provide clarification on the statistical significance of the results, particularly regarding the differences between T6SS1 and T6SS2 mutants. 

It appears that for one strain T6SS1 has an effect (linked to MIX domain-containing effectors?)

In Figure 8, consider presenting a core genome phylogeny and include bootstrap values for clarity.

Further clarification is needed regarding the rationale behind selecting strains for phylogeny in Figure 8 and the presence of homologs of CoVe2 and CoVe8 in all V. coralliilyticus genomes. Do they mean paralog/gene duplication? 

Consideration could be given to testing the deletion of candidate toxins in cellular assays, such as BMDM, for a more comprehensive understanding of their role in pathogenicity. If they tried but did not get an effect, what does it mean?

Discussion

Reevaluate the necessity of certain paragraphs, such as lines 355-364, for enhanced coherence and relevance.

Line 380: Ensure clarity in statements regarding the nature of Artemia infection assays. I don't believe V. coralliilyticus kills artemia externally, but rather after being ingested. Consequently, the bacteria are not exposed to ocean salt but instead to the stomach environment, which may not be nutrient-poor.

Lines 387-389. What is meant here? Is it suggesting that V. coralliilyticus could be a human pathogen? This assertion seems somewhat speculative and should not be the concluding statement. However, I find it intriguing that they conducted cellular assays with mouse cells, considering that Vibrio species from marine environments (non-human pathogen) typically struggle in salt-free media. 

Provide a stronger conclusion that encapsulates the main findings and implications of the study.

Reviewer #3: 

The authors report studies on the role of the Type 6 Secretion Systems (T6SSs) of the bacterial organism Vibrio coralliilyticus (Vcor) in assays for anrti-bacterial, anti-eukaryotic phagocytic cells, and anti-higher organisms (Artemia salina or brine shrimp). Because Vcor is a significant pathogen of both coral and shellfish, this study is potentially of high importance to understanding and potentially controlling this aquatic bacterial pathogen. 

The investigators performed studies to address whether the expression of two conserved ("omipresent') T6SSs of Vcor were regulated by growth temperature and found that indeed these were induced by elevated temperatures --- conditions that make Vcor particularly pathogenic for the model organism Artemia salina. This result is important because previous studies have proposed that elevated ocean temperatures modulate the virulence of this organism particularly for coral in the context of bleaching disease. 

The investigators go on to convincingly show that the T6SS1 has a major role as an antibacterial weapon using standard co-incubation on agar assays. They then show that the T6SS2 can elicit toxicity in both shrimp and in bone marrow derived macrophages from mice. These anti-eukaryotic effects of the T6SS are relatively rare in the field and thus these are important new results. The toxicity observed also correlated with loss of viability of the shrimp in their aquatic animal model. Using state-of-the-art proteomics the authors also determined the array of proteins that were likely secreted by the T6SS machines in Vcor. These included the usual structural components like Hcp, VgrG, and PAAR proteins but also a variety of likely effector proteins based on their homology to T6SS effector in other bacterial systems. Several of these proteins also did not display homology to known effectors and thus may indeed be novel toxins for prokaryotic or eukaryotic cells. 

I found the manuscript well-written and now reasonable to accept for publication in PLOS Biology. It addresses an important problem in microbiology -- how rising sea temperatures might affect the fitness and virulence of aquatic bacterial that in turn can then cause major economic and ecological damage globally.

---

## [Editor Report · Decision Letter 2]

27 Jun 2024

Dear Dr Salomon,

Thank you for your patience while we considered your revised manuscript "A T6SS in the coral pathogen Vibrio coralliilyticus secretes an arsenal of anti-eukaryotic effectors and contributes to virulence" for publication as a Research Article at PLOS Biology. This revised version of your manuscript has been evaluated by the PLOS Biology editors, the Academic Editor.

Based on our Academic Editor's assessment of your revision, we are likely to accept this manuscript for publication. Please also make sure to address the following data and other policy-related requests.

a) We routinely suggest changes to titles to ensure maximum accessibility for a broad, non-specialist readership, and to ensure they reflect the contents of the paper. In this case, we would suggest a minor edit to the title, as follows. Please ensure you change both the manuscript file and the online submission system, as they need to match for final acceptance.

"The coral pathogen Vibrio coralliilyticus uses a T6SS to secrete a group of novel anti-eukaryotic effectors that contribute to virulence"

Please supply the numerical values either in the a supplementary file or as a permanent DOI’d deposition for the following figures:

Figure 3ABC, 4ABCD, 5, 6, S3B, S4

c) Please cite the location of the data clearly in all relevant main and supplementary Figure legends, e.g. “The data underlying this Figure can be found in S1 Data” or “The data underlying this Figure can be found in https://doi.org/10.5281/zenodo.XXXXX”

d) Please provide the tree file for Figure 8

e) We thank you for providing the uncropped and minimally adjusted images supporting all blot and gel results reported. However please note that the labelling fin the raw images files, seems to be wrong in regard to the supplementary figures. Raw gels for S3A, S6, and S7, seemed to be now labelled as S1A, S3, and S4.

f) Please ensure that your Data Statement in the submission system accurately describes where your data can be found and is in final format, as it will be published as written there.

g) Per journal policy, if you have generated any custom code during the curse of this investigation, please make it available without restrictions upon publication. Please ensure that the code is sufficiently well documented and reusable, and that your Data Statement in the Editorial Manager submission system accurately describes where your code can be found.

We expect to receive your revised manuscript within two weeks. 

*Published Peer Review History*

*Press*

Sincerely,

Melissa

Melissa Vazquez Hernandez, Ph.D.

Associate Editor

PLOS Biology

---

## [Editor Report · Decision Letter 3]

3 Jul 2024

Dear Dr Salomon,

Thank you for the submission of your revised Research Article "The coral pathogen Vibrio coralliilyticus uses a T6SS to secrete a group of novel anti-eukaryotic effectors that contribute to virulence" for publication in PLOS Biology. On behalf of my colleagues and the Academic Editor, [**AE Name**], I am pleased to say that we can in principle accept your manuscript for publication, provided you address any remaining formatting and reporting issues. These will be detailed in an email you should receive within 2-3 business days from our colleagues in the journal operations team; no action is required from you until then. Please note that we will not be able to formally accept your manuscript and schedule it for publication until you have completed any requested changes.

PRESS

Sincerely, 

Melissa

Melissa Vazquez Hernandez, Ph.D., Ph.D.

Associate Editor

PLOS Biology
